# Brain Tuberculosis: An Odyssey through Time to Understand This Pathology

**DOI:** 10.3390/pathogens12081026

**Published:** 2023-08-09

**Authors:** Raluca Elena Patrascu, Andrei Ionut Cucu, Claudia Florida Costea, Mihaela Cosman, Laurentiu Andrei Blaj, Adriana Hristea

**Affiliations:** 1National Institute for Infectious Diseases Prof. Dr. Matei Bals, 021105 Bucharest, Romania; raluca.jipa1@drd.umfcd.ro (R.E.P.); adriana.hristea@umfcd.ro (A.H.); 2Infectious Diseases Department, Faculty of Medicine, University of Medicine and Pharmacy Carol Davila, 050474 Bucharest, Romania; 3Faculty of Medicine and Biological Sciences, University Stefan cel Mare of Suceava, 720229 Suceava, Romania; 4Emergency Clinical Hospital Prof. Dr. Nicolae Oblu, 700309 Iasi, Romania; claudia.costea@umfiasi.ro (C.F.C.); laurentiu-andrei.blaj@d.umfiasi.ro (L.A.B.); 5Department of Ophthalmology, University of Medicine and Pharmacy Grigore T. Popa Iasi, 700115 Iasi, Romania; 6Emergency County Hospital Braila, 810303 Braila, Romania; mihaelacosman@yahoo.com

**Keywords:** neurotuberculosis, meningitis, CNS tuberculosis, brain tuberculosis, tubercle bacillus

## Abstract

Tuberculosis is a contagious disease that has been a concern for humanity throughout history, being recognized and referred to as the white plague. Since ancient times, starting with Hippocrates and Galen of Pergamon, doctors and scientists have attempted to understand the pathogenesis of tuberculosis and its manifestations in the brain. If, in the medieval period, it was believed that only the touch of a king could cure the disease, it was only in the early 17th and 18th centuries that the first descriptions of tuberculous meningitis and the first clinico-pathological correlations began to emerge. While the understanding of neurotuberculosis progressed slowly, it was only after the discovery of the pathogenic agent in the late 19th century that there was an upward curve in the occurrence of treatment methods. This review aims to embark on an odyssey through the centuries, from ancient Egypt to the modern era, and explore the key moments that have contributed to the emergence of a new era of hope in the history of neurotuberculosis. Understanding the history of treatment methods against this disease, from empirical and primitive ones to the emergence of new drugs used in multi-drug-resistant tuberculosis, leads us, once again, to realize the significant contribution of science and medicine in treating a disease that was considered incurable not long ago.

## 1. Introduction

Infections of the nervous system have existed since ancient times, and the Egyptian mummies bear the stigmata of these diseases. Tuberculosis (TB) is an infectious disease caused by *Mycobacterium tuberculosis*, a bacteria with very ancient origins, which has survived for over 70,000 years [1,2]. TB is older than the human race, and different species within the genus *Mycobacterium* have historically caused diseases in fish, reptiles, birds, and mammals [3,4].

The purpose of this article is to make an exhaustive journey through history, from the earliest records regarding TB in antiquity to the ways in which treatment methods against this disease have evolved: from primitive methods to the emergence of newly drugs administered in multidrug-resistant TB. It is worth noting that these treatment methods have evolved in parallel with humanity’s understanding of this disease’s origin, transmission, and clinical and pathological manifestations. Let us enjoy this journey!

## 2. Tuberculosis in Antiquity

Archaeological explorations have identified Egyptian mummies dating back to 2400 BC with typical skeletal deformities of TB [5], some of which are spectacularly depicted in early Egyptian art [6,7] (Figure 1A). 

Although there is no evidence of descriptions of tuberculous lesions in Egyptian papyri, the earliest written documents mentioning TB date back to 3300–2300 BC and come from China and India [8,9]. Spinal column disorders resembling Pott’s deformities have also been identified in Peruvian mummies, suggesting that the disease was present in the South American continent prior to the first European colonization [10,11,12].

TB was well-known in ancient Greece, where it was called *phthisis* (consumption), and Hippocrates (c. 460 BC–c. 370 BC) considered it a fatal disease, especially for the young [2]. Isocrates (436–338 BC) was the first author to consider TB as an infectious disease, and Aristotle (384–322 BC) regarded it as contagious [2,13]. Later, Galen of Pergamon (129-c. 216 AD) mentioned the clinical signs and symptoms of TB, which included bloody sputum, sweating, and fever, and recommended treatments such as consuming milk, fresh air, and sea voyages [14,15,16]. After the fall of the Roman Empire, between the 8th and 19th centuries, TB spread throughout the entire European continent [17]. 

Hippocrates was among the first to describe the clinical presentation of brain TB, characterizing it as dominated by headaches, fever, and altered consciousness [18]. Additionally, Hippocrates was the first to observe tubercles (*phymata*) in the tissues of pigs, sheep, or cattle [19]. Analogues of these tubercles could not be seen in humans, because in ancient Greece, human dissections were uncommon and unacceptable due to superstitions related to the desecration of human bodies, and dissections were only allowed for animals, particularly monkeys [20].

Furthermore, the Hippocratic school considered pulmonary *phthisis* (TB) as a hereditary disease rather than an infectious one [19]. On the other hand, Aristotle regarded it as a contagious disease, describing *scrofula*, a characteristic skin lesion in tuberculous (*phthisic*) pigs, which he considered to be a contagious lesion [21].

## 3. Tuberculosis in the Middle Ages: The Royal Healing Touch

From the time of Galen until the 8th century, there were no significant developments in the history of TB [22]. During the medieval period, *scrofula* was known as “the King’s Evil” because it was believed to be curable through the touch of the king, a practice particularly prevalent in France and England [22] (Figure 1B).

This belief originated with King Clovis of France (487–511) (Figure 1B), and later, other European monarchs such as Edward the Confessor, Philip I of France and Robert the Pious reinforced this belief [23,24]. 

Later, Richard Wiseman (1622–1676), the personal surgeon of King Charles II (1630–1685), wrote in the fourth book of his 1672 *Treatise on the King’s Evil* that the monarch touched 92,102 subjects during his reign of 25 years [25]. This custom, which began in the 12th century, persisted until the 18th century [26,27,28]. On the other hand, in Islamic medicine, physicians such as Avicenna, Rhazes, and Albucasis made pragmatic contributions to the understanding of this disease, describing its pathology, symptoms, and predisposing factors [22].

In 1679, the Dutch physician Franciscus Sylvius (1614–1672) used the term *tubercles* to describe the lesions of lung phthisis [19,22]. In his work *Opera Medica*, he referred to them as *tubercula glandulosa* (glandulous tubercles), believing that they could progress into ulcers, abscesses or empyema [19]. Furthermore, Sylvius was the one who described the association between *phthisis* and *scrofula*, the disease affecting the lymph glands of the neck [19].

## 4. The First Clinical Observations from the 17th Century

Regarding the understanding of the pathogenesis of cerebral TB, the 17th century was dominated by the work *Phthisiologia*, published in 1689 by the English physician Richard Morton (1637–1698) [19,29,30] (Figure 2A). In his work (Figure 2B), Morton described the pulmonary pathology of this disease as well as its severity in young individuals, referring to it as “the consumption of young men, that are in the flower of their age” [29,30].

In the same century, Thomas Willis (1621–1675), in his famous work on the anatomy of the nervous system *De Anima Brutorum*, made several observations about brain TB: “nec minus a phlegmone et abcessu quam hujas modi meningitis et tuberculis, cephalgiae lethales et incurabiles oriuntur (sometimes the headaches, fatal and incurable, follow abscesses and swellings of the envelopes of the brain, as well as plaques and tubercles of theses membranes)” [31]. In his descriptions, the death of the patient with brain TB occurred after a state of lethargy and prolonged fever. Furthermore, during the postmortem examinations of these patients, Willis observed inflammation of the meninges, dilation of the cerebral ventricles, and tuberculomas compressing the brain [32,33]. It seems that the connection between serous effusion and TB was made by Thomas Willis. In such cases, the most commonly proposed treatment was bloodletting [33].

## 5. The 18th Century: The First Description of Tuberculous Meningitis and the First Clinical–Pathological Correlations

At the beginning of the 18th century, the first comprehensive clinical descriptions of brain TB were made by Scottish physicians Andrew Saint-Clair (?–1728) [34] and John Paisley (c. 1679/1729–1740) [35]. Paisley described the case of a 7-year-old child who, after a prolonged state of lethargy, fever and coma, passed away. During the postmortem examination, Paisley reported: “in dissecting the brain, I found the ventricles had been much distended and enlarged by the water; the plexus choroeides were hard and scirrhous with a great number of small hydatide (as I supposed) lying along the rows, whose coats were exceeding tender and burst upon the least touch. They remember the lymphaticks delineated in the fifth of Dr Rudley’s anatomy of the brain” [33,35].

Old descriptions of extrapulmonary manifestations of TB were written in 1779, when Percival Pott (1714–1788) described Pott’s disease of the spine [36]; in 1790, when Marc-Antoine Petit (1766–1811) described tuberculous laryngitis [37]; in 1796, when Robert Willan (1757–1812) described erythema nodosum [38]; and in 1839–1841, when Pierre-François Olive Rayer (1793–1867) provided the first complete description of renal TB [39]. They were followed by Carl Rokitansky (1804–1878), who in 1842–1846 described intestinal TB [40], and by Paul Brouardel (1837–1906), who described genital TB in women in 1895 [41,42]. In 1718, the French surgeon and anatomist Jean-Louis Petit (1674–1750) was the first to describe TB of the mastoid [33,43,44]. It was only in 1883 that Franz Eschle became the first to demonstrate the presence of the tubercle bacillus in the secretion from the middle ear in patients with TB [45], one year after Koch discovered the tubercle bacillus [43]. However, the involvement of the nervous system by the tubercle bacillus was described quite early by Sir Robert Whytt (1714–1766) in 1768 [46].

Professor of medicine at Edinburgh and physician to George III in Scotland, Sir Robert Whytt [47] (Figure 3A) was the first to describe *tuberculous meningitis*, although his work was published several years after his death, in 1768, in his famous book *Observations on the Dropsy in the Brain (1768)* [46] (Figure 3B). 

In this work, based on 20 cases, Whytt accurately described the signs and symptoms of a condition that would later be recognized as tuberculous meningitis [48]. With great precision, Whytt classified the clinical picture of dropsy of the brain into several stages (Figure 3C,D). The first stage was characterized by drowsiness, vomiting, headache, photophobia, and a rapid and regular pulse; the second stage involved the continuation of the symptoms from the first stage, along with a slower and irregular pulse and double vision. Towards the end of the second stage, children became “delirious and frightened”. The third stage was characterized by an increase in pulse rate, which occurred 5–7 days before death. Furthermore, in this stage, the child became drowsy and comatose, developed eyelid paralysis, and their pupils remained dilated in the greatest light [46,48].

During the postmortem examinations of 10 patients, Whytt also observed a clear and thin liquid in the anterior ventricles of all of them, in the third or fourth ventricles, but never between the dura mater and the brain. The fluid did not coagulate upon heating [46,48]. Whytt considered that “fever from water of the brain is easily distinguished from others by attending to the whole cause of the disease and particularly to the pulse (…) at first quick (…) slow and irregular (…) and lastly acquires a greater frequency than ever. Besides, the screaming, squinting, and dilatation of the pupils rarely occur in other fevers” [46,48].

## 6. The 19th Century: Bringing to Light the Pathogenesis of Brain Tuberculosis and the Discovery of the Koch Bacillus

During the course of the Industrial Revolution in the 1780s in Europe, the incidence of TB further increased, reaching epidemic proportions and becoming the leading cause of death in the 18th century. In continental Europe, progress was made in understanding this disease, particularly from the late 18th century. While France had many researchers advancing the understanding of TB, the breakthrough in TB diagnosis came from the Germans, with Professor Robert Koch himself discovering the causative agent in 1882. This was followed immediately in the period from 1882 to 1888 by Franz Ziehl’s (1859–1926) and Friedrich Neelsen’s (1854–1898) report of the Ziehl–Neelsen staining technique, which aided in the identification of the bacillus. Later, in 1895, Wilhelm Conrad Röentgen (1845–1923) discovered Röentgen’s X-ray, which further advanced the diagnosis of TB [22].

### 6.1. The First Theories of Inflammation of the Pia Mater and Arachnoid Membrane

In 1790, Edward Ford proposed one of the early theories on the pathogenesis of neurotuberculosis, suggesting that the hydrocephalus observed in tuberculous meningitis is either a result of inflammation of the pia mater or a consequence of scirrhus induration (TB) of the brain and cerebellum [49]. Following Edward Ford’s theory, most savants began to consider inflammation as the primary factor in tuberculous meningitis, with effusion being a secondary factor. However, there were divergent opinions regarding the initial localization of the disease and its subsequent extension.

Two decades later, Leopold Anton Goelis (1764–1827) believed that the arachnoid membrane was the site of the pathological process, while Jean-Francois Coindet (1774–1834), in the same year, considered the ventricular system to be the site of the pathological process. Later, John Abercrombie (1781–1844) placed the lesion at the level of the brain, while Jean-Louis Brachet (1789–1858) placed at the level of the lymphatics, and Piorry, in 1822, similar to his predecessor Goelis, placed it at the level of the arachnoid membrane. Bichat completed this theory by mentioning “that the tissues belonging to the brain, by the arachnoid, to the lungs by the pleura, to the abdominal viscera by the peritoneum, it matters not which, may inflame all over in the same manner. Either the hydropsy comes on uniformly or it is subject to a species of eruption miliary-like and whitish, which has not been mentioned, I believe, and which nevertheless merits great consideration” [31].

In 1820, the French scientists Louis Martinet (1795–1875) and Alexandre Parent du Châtelet (1790–1835) presented a dissertation to the Royal French Academy of Sciences on the inflammation of the arachnoid membrane, stating that “tubercles, due to their development, may sometimes cause an inflammation of the arachnoid”. The two reached this conclusion after analyzing 140 cases, describing a new condition they called *arachnitis*, which they believed could be caused by various microbes that they were unable to identify at the time [33,50].

Five years later, in 1825, the French physician Louis Senn from Geneva (1799–1873) described a chronic form of tuberculous basal meningitis occurring in children [51]. Following autopsies on children with TB, Senn observed miliary disease, pulmonary cavitation, and brain tubercles. Additionally, he identified “little yellowish and lenticular plaques” in the brain, typically located near blood vessels, and the meninges, especially those at the base of the brain, were thick, opaque, and filled with granulations [18,51]. One year later, the English physician Robert Hooper (1773–1835) (Figure 4A) depicted a purulent basilar meningitis of tuberculous origin in his atlas, *The Morbid Anatomy of the Human Brain (1826)* [52]. The work contained a section dedicated to inflammations of the pia mater and arachnoid, with dramatic images showing enlarged and turgid blood vessels of the pia mater and significant amounts of “puriform albumen” [52] (Figure 4B).

### 6.2. Granulomatous Meningitis

Undoubtedly, the cause of the disease needed to be sought, as it had long been associated with *scrofula*. In 1819, Guibert and Charpentier used the term *granulations* to describe the condition of the meninges, and Louis Senn even described a granular form of the disease. In 1827, the French physician and botanist Louis Benoît Guersant (1777–1848) used the term *meningitis granuleuse* and reported the presence of TB in other organs, not just in the brain. Later, in 1830, Papavoine L.N. introduced the term *arachnitis tuberculeuse*, distinguishing two forms: granulomatous and plaque-like [53,54]. Papavoine also observed that TB precedes effusion and that there is a coincidence with the presence of TB in other organs [31]. Five years later, Papavoine wrote three monographs in which he demonstrated that the granulations were actually tubercles, that these granulations in the brain were identical to those in other serous membranes, that they only appear in the brain when other organs are affected, and that acute hydrocephalus is caused by TB [31]. Later, in 1839, Guersent proposed a classification of meningitis into tuberculous meningitis and non-tuberculous (simple) meningitis [55,56].

The French physician Étienne Rufz de Lavison (1805–1884), a former resident of Guersant, also published 10 cases of dropsy of the ventricles, in which he noted miliary lesions or tubercles in various parts of the brain. He considered these lesions to be the cause of meningeal inflammation, leading to an increase in the volume of fluid in the cerebral ventricles [57]. In addition to the miliary lesions and tubercles in the meninges and the dilatation of the cerebral ventricles, Rufz also observed the presence of constant tuberculous pulmonary lesions [58].

Similar cases were also described and published on the American continent by the American physician William Wood Gerhard (1809–1872) from Philadelphia. He described 10 cases from his own experience and another 20 cases collected from his colleagues at the Hôpital des Enfants Malades in Paris [59]. Gerhard reported the presence of headache, seizures, vomiting, and other focal neurological signs, including muscle paralysis or pupil abnormalities. Additionally, Gerhard observed that these neurological signs could occur even in the absence of hydrocephalus [18,59]. A few years later, in 1834, the German physician Johann Lukas Schönlein (1793–1864) introduced the term *tuberculosis (tuberkulose)*, derived from the term *tubercles* [60].

In 1837, the French physician Victor-Mathurin Le Diberder (1810–1891), a disciple of the renowned surgeon and anatomist Alfred Velpeau (1795–1867), defended his thesis in which he mentioned his observations that the granulations were found “most often in the base of the brain and extended in the Sylvian fissure, along the middle cerebral artery” [61]. Furthermore, Le Diberder correctly attributed the ocular involvement and coma to the increased volume of fluid [61]. His compatriot, the pediatrician François Louis Isidore Valleix (1807–1855), was also convinced of the importance and novelty of this subject. In his work *De la méningite tuberculeuse chez l’adulte*, Valleix observed that all patients, whether adults or children, in addition to characteristic pulmonary lesions, frequently had peritoneal granulations and meningeal tubercles [62]. He also believed that death occurred as a result of tuberculous meningitis. Thus, in the 1830s–1840s, physicians reached an initial conclusion that dropsy of the ventricles occurred due to tuberculous infections of the central nervous system [33].

## 7. The First Clinical–Pathological Correlations

An important moment in the history of brain TB occurred in 1881, when Carl Wernicke (1848–1905) performed the first sterile ventricular procedure and external drainage of cerebrospinal fluid [63]. He was followed a few years later by the German physician Heinrich Quincke (1842–1922), who introduced lumbar puncture for both diagnostic and therapeutic purposes [64,65].

Another important moment was the invention of the stethoscope by the French physician René Théophile Hyacinthe Laënnec (1781–1826) at the Necker–Enfants Malades Hospital (Figure 5), which proved to play a major role in the clinical diagnosis of TB. 

Moreover, Laënnec correlated pulmonary auscultation with observations during postmortem studies, where he identified both pulmonary and extrapulmonary tuberculous lesions. He provided a detailed description of how tubercles initially appear in the lungs as miliary TB (resembling millet seeds) and progress to tubercles containing caseous material (cheese-like). Laënnec also described the transformation of tubercles into pus, the formation of cavities and empyema, as well as extrapulmonary phthitic tubercles in other organs, including the meninges and brain. His work elucidated the pathogenesis of TB and unified the concept of pulmonary and extrapulmonary disease [22,66,67,68]. Laënnec stated that there is no organ unaffected by these tubercles, mentioning the brain, cerebellum, cranial bones, and vertebral column [22,69]. He postulated that all tubercular phenomena, such as *scrofula*, miliary TB, or *phthisis*, constitute a single disease [22]. In 1826, in Kerlouanec (France), Laënnec himself succumbed to TB at the young age of 45 [66]. It is likely that he contracted the disease during his work dissecting cadavers, where Laënnec was exposed to numerous finger cuts [67,70].

Laënnec was followed by Jean-Marc-Gaspard Itard (1774–1838), who, after studying the Greek and Arab authors who wrote about hydrocephalus, provided information about the natural progression of the disease and was one of the first authors to discuss its etiology: “more modern research on pathological anatomy, due in large part to Laënnec, has shown tuberculous granulations of the substance of the brain and cerebellum, in the layers of the optical nerves, and even in the thickness of the meninges”. He also described the performance of a chemical analysis of the abundant fluid present in the cerebral ventricular system, and his conclusion was that the effusion is not the disease itself but only its result [33,71]. Later, in 1865, the French physician Georges Simonis Empis (1824–1913) introduced the term *la granulie* (miliary TB), which represented a clinical–pathological syndrome characterized by fever, granular meningitis, and acute hydrocephalus [72].

## 8. From the Cannulation Technique to the First Treatments against Tuberculosis

Several years later, in 1889, the English physician Walter Essex Wynter (1860–1945) described the technique of cannulation in patients with tuberculous meningitis, which had more therapeutic than diagnostic significance. Wynter made a small incision at L2, cut through the dura mater, and introduced a Southey tube with a rubber drain to drain the infected fluid and reduce pressure [73]. However, the one who definitively introduced the lumbar puncture technique was the German physician Heinrich Irenaeus Quincke (1842–1922) [64,65], who presented his personal experience in 1891 at the Conference of Internal Medicine in Wiesbaden, Germany. Quincke developed lumbar puncture initially for treatment and later for the bacteriological analysis of cerebrospinal fluid [64,65,74]. In the United States, the procedure was implemented around the same time at a children’s hospital by Arthur Howard Wentworth [75]. Also, in the same year, Walter Wynter and Charles Morton recommended therapeutic lumbar drainage for patients with tuberculous meningitis [76,77].

After 1882, the understanding of the pathogenic mechanisms of TB took a significant leap, when the German physician and microbiologist Robert Koch (1843–1910) (Figure 6A) identified the causative agent of TB [78] (Figure 6B).

On the evening of 24 March 1882, Robert Koch was delivering a presentation in front of the Berlin Physiological Society that would revolutionize medicine. He announced that he had succeeded in identifying the causative agent of TB, which was a rod-shaped bacterium, which he then called tubercle bacillus [79]. A few years later, in 1905, he was awarded the Nobel Prize in Physiology or Medicine for “his investigations and discoveries related to tuberculosis” [80].

In 1896, Adams reported the first tubercular abscess of the brain [81,82,83]. Later, in 1960, Thiébaut and Philippides emphasized the importance of differentiating tuberculous brain abscesses from cerebral tuberculoma [83,84]. However, there are few properly verified reports of tuberculous brain abscesses in the literature [83]. Three years prior to Adams’ report, Scottish surgeon William Macewen (1848–1924) (Figure 7A) recommended in his work *Pyogenic Disease of the Brain and Spinal Cord. Meningitis, Abscesses of the Brain, Infective Sinus Thrombosis (1893)* (Figure 7B) that, in cases of cerebral abscesses (Figure 7C), the abscess should be drained, and the underlying cause of the infection should be appropriately treated [85]. 

Macewen included a special subchapter entitled *Tuberculosis and carcinosis of the middle ear*, in which he described the clinic and pathogenesis of TB located in the middle ear. He also mentioned that these cases can be accompanied by tuberculous abscesses in the brain, though with a rapidly fatal character if the tubercles have spread, with the subsequent appearance of tubercular meningitis.

In the case of a 45-year-old patient who died of tubercular ulceration at the base of the skull, Macewen mentioned: “at a post-mortem examination of a man (C.P.O., 1880, aged 45 years, who was markedly tubercular, the lungs being filled with large caseating masses, the bones of the base of the skull were of a bluish appearance, from the contained masses of granulation tissues, showing through the white bone. So thin were these bones that pressure with the tip of the finger caused the internal table to crumble and to expose the granulation tissue between the two tables of the bone. This was especially apparent over the basi-sphenoid and basi-occipital, and both petrous and mastoid bones (…). There was a slight amount of pus in both middle ears, (…). Patient died from tubercular exhaustion, but had been doing business until a month before his death. The dura at various points at the base was eroded (…) and there was chronic tubercular lepto-meningitis” [85]. 

Macewen, in several chapters, also addressed middle ear infections and their complications, such as mastoiditis, leptomeningitis and cerebral or cerebellar abscesses (Figure 8). 

Subsequently, in the treatment of cerebral abscesses, Joseph King introduced marsupialization in 1924 [86], and Walter Dandy introduced aspiration in 1926 [87]. In 1928, Percy Sargent (1873–1933) considered the procedure of enucleation of the cerebral abscess [88], and in 1936, Clovis Vincent (1879–1947) recommended complete excision of the cerebral abscess [89].

In the 1930s, Howard McCordock (1895–1938) from Washington University School of Medicine and Arnold Rice Rich (1893–1968) from Johns Hopkins Medical School conducted significant pathological studies on tuberculous meningitis, leading to the recognition that the disease occurs as a result of the discharge of bacteria into the subarachnoid space from a caseous focus located either in the brain or in the cerebral meninges [90].

In 1942, Irish pediatrician Dorothy Stopford Price (1890–1954) provided a precise guide to the clinical presentation of cerebral TB [48]. Regarding tuberculous meningitis, she stated: “tuberculous meningitis in children is an acute and fatal disease appearing early in hematological spread, and early in the whole disease picture (…). The early supervention of the meningitis offers not time for wasting” [48,91]. Additionally, Price identified an acute condition called *terminal meningitis* that occurred in children with miliary or tertiary TB of the lung [91]. Price’s work was the key to the elimination of childhood TB in Ireland by introducing the Bacillus Calmette–Guérin vaccine [92].

## 9. The Beginnings of Tuberculosis Treatment

Throughout history, the treatment of TB has been more of a cause for frustration and helplessness, often consisting of a variety of herbal preparations, dietary changes, and climatic prescriptions [93]. However, with the discovery of penicillin in 1928 by Scottish bacteriologist Alexander Fleming (1881–1955) and sulfonamides in 1935 by German pathologist and chemist Gerhard Domagk (1895–1964), the age of chemotherapy and effective antimicrobial therapy against TB began (Table 1) [93].

Later, in 1943, in New Jersey, Selman Waksman (1888–1973) identified streptomycin, the antibiotic that was to strengthen the fight against the TB bacillus. Its intramuscular and intrathecal administration for a period of 3 months reduced the cell count below 20, leading to an improvement in the overall condition of the patient. As a result, between 1948 and 1953, half of the patients recovered under this treatment [94].

In the same year, Professor Jörgen Erik Lehmann (1898–1989) from Sahlgrenska University Hospital synthesized the para-amino salt of salicylic acid (PAS). Subsequently, due to the lack of streptomycin, the British Medical Research Council (BMRC) conducted one of the first randomized clinical trials comparing PAS and SM alone or in combination [95]. The results, published in 1950, demonstrated that this combination was more effective in both achieving cures and preventing the development of drug resistance.

In 1962, Georges Canetti (1911–1971) proposed the principle of two-phase chemotherapy [96], and in 1965, Denis Anthony Mitchison (1919–2018) recommended initial therapy with a combination of three drugs [97]. Starting in the 1960s [98] and 1970s [99,100], the BMRC trial introduced regulations regarding the duration of drug administration. Undoubtedly, these studies significantly shaped future treatment trials. 

In recent decades, drug-resistant TB (DR-TB) has emerged as the main challenge for the WHO Global TB Programme, as it increases the risk of disease relapse, treatment failure, and death [101]. Multidrug-resistant tuberculosis (MDR-TB) is defined as the appearance of MTB strains resistant to at least rifampicin and isoniazid, making these patients practically incurable using standard first-line TB drugs [102,103]. Additionally, in 2006, a new category named extensively drug-resistant (XDR) TB emerged, referring to MDR-TB strains that are resistant to fluoroquinolones and second-line injectable drugs [102]. MDR-TB has a high mortality rate, and due to the spread of this new form, there is a risk of tuberculosis becoming an incurable disease. These concerns have led to numerous studies in recent years that have evaluated the risk factors for MDR-TB, such as lung cavity, previously diagnosed TB, previous anti-TB therapy, and sputum acid-fast bacilli smear positivity [104,105,106], which are risk factors independent of the global context. Considering these aspects, special attention should be given to these categories of patients.

The constant and ongoing spread of MDR-TB represents a challenge and an urgent situation for TB control globally [102]. In this regard, scientists have made tremendous efforts to identify new anti-tuberculosis drugs.

In 2005, Andries et al. [107] discovered bedaquiline, which was later approved by the US FDA for the treatment of MDR-TB in 2012. Subsequently, delamanid emerged, which proved its effectiveness first in preclinical studies conducted in vivo and in vitro [108,109] and later in multinational clinical trials [110].

In 2013, delamanid was approved by the European Medicines Agency. Another significant moment in TB treatment occurred in 2019, when the first 6-month regimen was approved for the treatment of MDR and XDR TB, consisting of bedaquiline, pretomanid and linezolid [111]. Currently, the Working Group on New TB Drugs is trying to accelerate the discovery and development of new drugs for TB treatment [112].

## 10. Conclusions

Neurotuberculosis has been known and observed since the time of ancient Egypt and Greece, and European scientists attempted to understand this condition. Despite glimpses into its pathological mechanisms and clinical characteristics, its cause remained unknown until the end of the 19th century, when the TB bacillus was discovered. Subsequently, the advent of antibiotics and their use in the treatment of TB starting in the 1930s marked a new era of hope in the history of TB.

## Figures and Tables

**Figure 1 pathogens-12-01026-f001:**
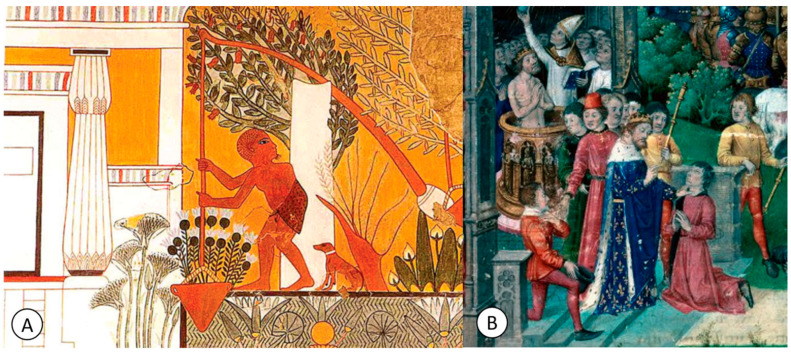
(**A**) Image from the tomb of the building master Ipwy (19th dynasty, Deir el-Medina) which depicts a gardener with spine tuberculosis (with possible Pott’s disease) (public domain). (**B**) A 15th century image representing King Clovis I touching for scrofula (public domain).

**Figure 2 pathogens-12-01026-f002:**
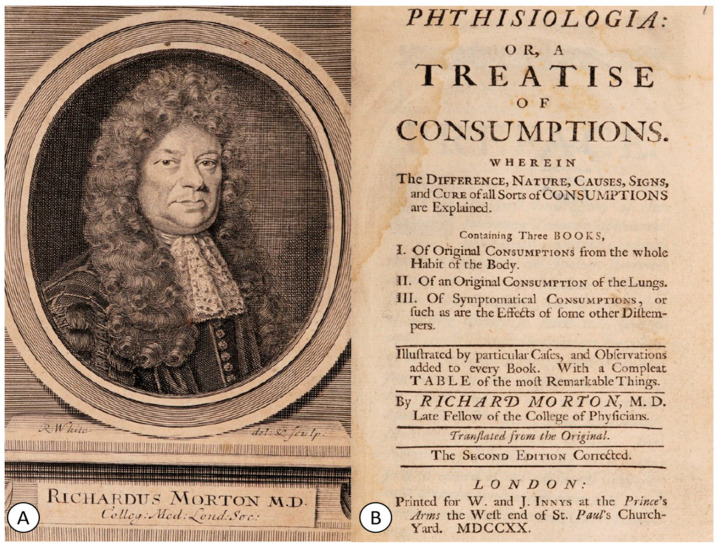
(**A**) Portrait of Richard Morton (1637–1698) represented in his book. (**B**) The title page of his book *Phthisiologia: or, a Treatise of Consumptions*, printed for W. And J. Innys, London (1720) (public domain).

**Figure 3 pathogens-12-01026-f003:**
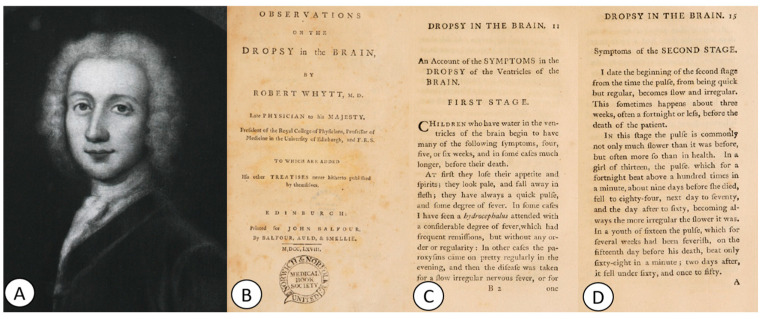
(**A**) Portrait of Robert Whytt (1714–1766). (**B**) The title page of his work *Observations on the Dropsy in the Brain*, printed for John Balfour by Balfour, Auld & Smelie, Edinburgh (1768). (**C**) Original description of the first stage and (**D**) the second stage of tuberculous meningitis (first pages) (public domain).

**Figure 4 pathogens-12-01026-f004:**
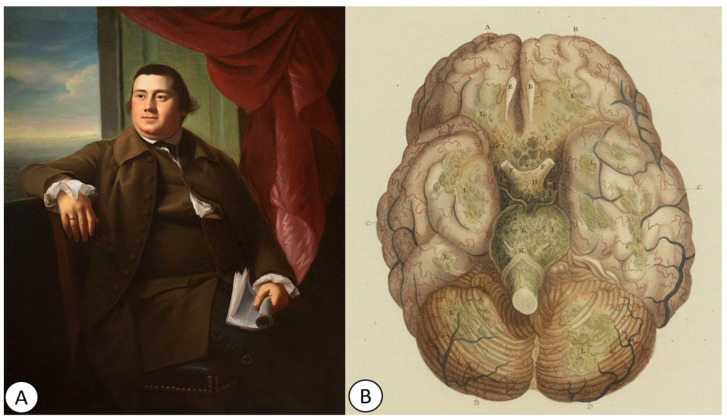
(**A**) Portrait of Robert Hooper (c. 1770/1772) by John Singleton Copley (1738–1815), Smithsonian American Art Museum. (**B**) Illustration of a purulent basilar meningitis in his atlas *The Morbid Anatomy of the Human Brain* (1826) by Robert Hooper: “Plate IV. INFLAMMATION OF THE PIA MATER and TUNICA ARACHNOIDES: The cerebrum and cerebellum so placed as to bring the whole of the base or under surface of the brain into view. The same diseases appearances are seen in this as in the former plate, but to a much greater extent: a puriform albumen is secreted on the surface of the lobes of the cerebrum, on that of the cerebellum, and over the whole of the medulla oblongata” (public domain).

**Figure 5 pathogens-12-01026-f005:**
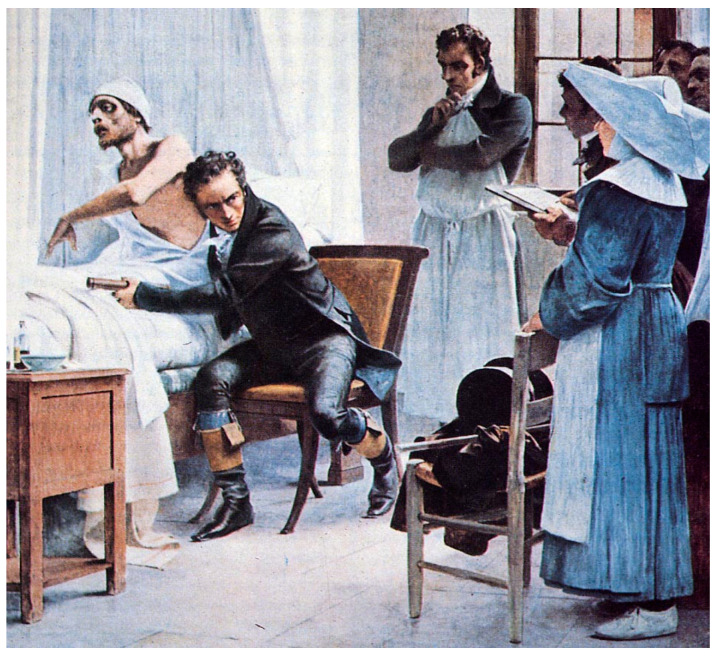
“Laënnec à l’hôpital Necker ausculte un phtisique devant ses élèves” (Laënnec examines a consumptive patient with a stethoscope in front of his students at the Necker Hospital), painting by Théobald Chartran (1816) (public domain).

**Figure 6 pathogens-12-01026-f006:**
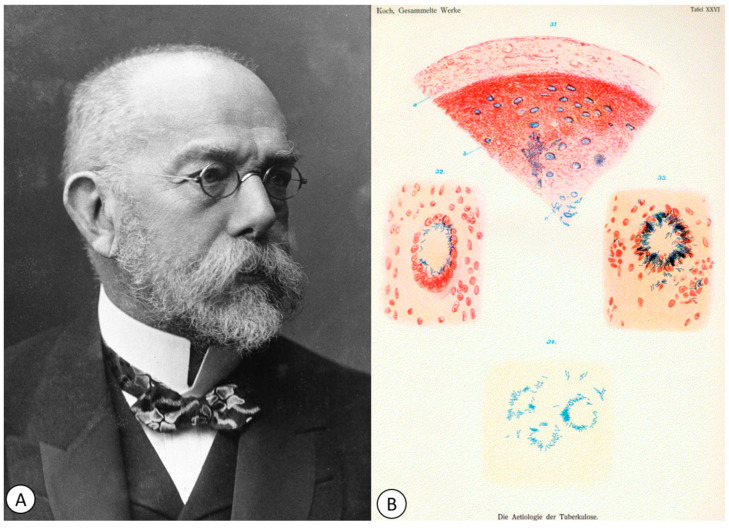
(**A**) Portrait of Professor Robert Koch (1843–1910), published in 1907 in Les Prix Nobel. (**B**) Koch’s drawing of tuberculosis bacilli in 1882 (from *Die Ätiologie der Tuberkulose*) (public domain).

**Figure 7 pathogens-12-01026-f007:**
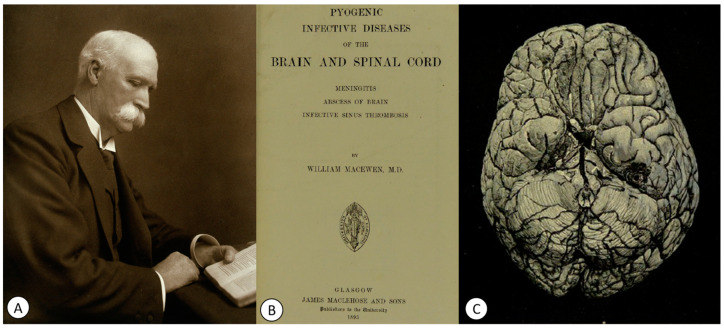
(**A**) Photograph of William Macewen (c. 1910). (**B**) The title page of his book, *Pyogenic Disease of the Brain and Spinal Cord. Meningitis, Abscesses of the Brain, Infective Sinus Thrombosis* (1893). (**C**) Figure 37 from his book: “Abscess in left temporo-sphenoidal lobe. Aperture in base of brain, connected with abscess in temporo-sphenoidal lobe, and with granulation mass springing from the membranes and causing an indentation on cerebral surface. The tegment antri and tympani were eroded. From photograph” (public domain).

**Figure 8 pathogens-12-01026-f008:**
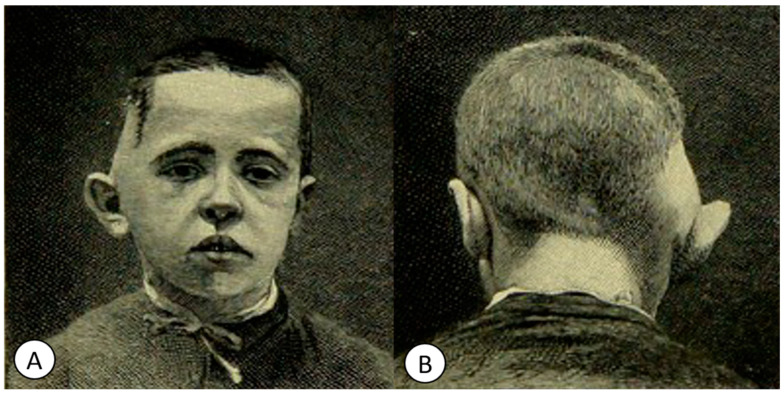
Cases of acute sub-periosteal squamo-mastoid abscesses, presented in the book *Pyogenic Disease of the Brain and Spinal Cord. Meningitis, Abscesses of the Brain, Infective Sinus Thrombosis*. (**A**) “Front view appearances presented in acute sub-periosteal squamo-mastoid abcess. The forward displacement of the ear is well illustrated (Case V)”. (**B**) “Appearance presented in acute sub-periosteal squamo-mastoid abscess of large size (case VI)” (public domain).

**Table 1 pathogens-12-01026-t001:** The main contributors to the understanding of the pathogenesis, clinical characteristics and treatment of tuberculosis.

Year	The Name of Contributor	The Discovery Achieved
2400 BC	Egyptian mummies with spinal deformities typical of TB	
400–300 BC	Hippocrates	the clinical presentation of brain tuberculosis
1679	Franciscus Sylvius (1614–1672)	uses the term *tubercles*
1689	Richard Morton (1637–1698)	*Phthisiologia*
1768	Robert Whytt (1714–1766)	describes tuberculous meningitis for the first time
1779	Percival Pott (1714–1788)	Pott’s disease of the spine
1819	Rene T H Laënnec (1781–1823)	describes the *tubercles*, invents the stethoscope
1827	Louis Benoît Guersant (1777–1848)	describes *meningitis granuleuse*
1834	Johann Lukas Schonlein (1793–1864	introduces the term *tuberculosis* (tuberkulose)
1882	Robert Koch (1843–1910)	discovers *tubercle bacillus*, the pathogen that causes TB
1889	Walter Essex Wynter (1860–1945)	describes the cannulation technique in patients with tuberculous meningitis
1890	Friedr Neelson (1854–1898) Franz Ziehl (1859–1926)	Ziehl–Neelsen acid-fast staining
1891	Heinrich Quinke (1842–1922)	lumbar puncture
1895	Wilhelm Conrad Röentgen (1845–1923)	discovers X-rays
1896	Adam S	reports the first tubercular abscess of the brain
1943	Selman Waxman (1886–1973)	discovers streptomycin (SM)
1944	Jörgen Erik Lehman (1898–1989)	discovers the para-amino salt of salicylic acid (PAS)
1948	BMRC *, first randomized trial: SM versus PAS versus SM/PAS	
1962	Georges Canetti (1911–1971)	proposes the principle of two-phase chemotherapy
1965	Denis Anthony Mitchison (1919–2018)	recommends initial therapy with a combination of three drugs
2005	Andries et al.	discovers bedaquiline
2013	WHO	*Monoresistance*—resistance to one first-line anti-TB drug alone *Polydrug resistance*—resistance to more than one first-line anti-TB drug other than both isoniazid and rifampicin *Multidrug resistance* (MDR)—resistance to at least both isoniazid and rifampicin *Rifampicin* resistance (RR) *Extensive drug resistance* (XDR)—resistance to any fluoroquinolone and at least one of three second-line injectable drugs
2013	European Medicines Agency approves the use of delamanid	
2019	The first 6-month regimen was approved for the treatment of MDR and XDR TB: bedaquiline, pretomanid and linezolid	
Present	Working Group on New TB Drugs	

* BMRC—British Medical Research Council, PAS—para-amino salt of salicylic acid, SM—streptomycin, TB—tuberculosis, WHO—World Health Organization.

## Data Availability

In order to have access to the database, please contact the correspondent author.

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
