# Peer review of "Brain Tuberculosis: An Odyssey through Time to Understand This Pathology"

_pathogens, 2023, doi:10.3390/pathogens12081026_

Round 1

Reviewer 1 Report

Thank You for your contribution.  This is an interesting piece and really does take the reader on a fascinating historical journey of CNS TB.  I think this will potentially be widely read to anyone involved in the care of patients with TB or in TB research.  The story flows well and the figures are an important addition to the piece.

There are a few typographical errors and some sentences which were repeated.  The piece would benefit from another proof read. Overall, well written.

Author Response

Response to Reviewer 1 Comments

Comments and Suggestions for Authors

Thank You for your contribution.  This is an interesting piece and really does take the reader on a fascinating historical journey of CNS TB.  I think this will potentially be widely read to anyone involved in the care of patients with TB or in TB research.  The story flows well and the figures are an important addition to the piece.

Comments on the Quality of English Language

There are a few typographical errors and some sentences which were repeated.  The piece would benefit from another proof read. Overall, well written.

Response 1: Please provide your response for Point 1. (in red)

The authors would like to thank the reviewer for his words of appreciation for this article. We are grateful if we managed to take the reader on an interesting and cursive journey of CNS TB and that this article can be read widely by all those involved in the care of TB patients.

We corrected the typos (marked them in red) and we have deleted the sentences that are repeated

Thank you once again for the words of appreciation that honor us.

Thany you once again for the extremely valuable comments.

Reviewer 2 Report

Your work on the historical course of brain tuberculosis is truly commendable. The thoroughness with which you have researched this complex subject is a testament to your dedication and expertise in the field. By illuminating the often overlooked historical context of this disease, your manuscript provides invaluable insights that enrich our understanding of tuberculosis as a whole. The choice of topics and the logical structure of your report make it a compelling and exciting read. Your efforts to cover a broad period from antiquity to the present demonstrate a great commitment to presenting a holistic view of this subject.
Some general comments:
1. Abstract: It would be beneficial to reword the abstract to align it more closely with the text, especially with regard to the goal and conclusion.
2. Introduction:  Consider explicitly stating in the introduction the purpose of the article, the nature of the study conducted, and its overall significance to provide clear guidance to the reader.

3. The TB treatment: Given the significant advances that have been made over the past five decades, I recommend that you update the manuscript with the most recent findings, particularly with respect to the treatment of multidrug-resistant forms TB. Including these advances in both the main text and Table 1 would increase the relevance and impact of your work.

I hope you find these comments helpful in further refining your manuscript. Your commitment to submitting a comprehensive and academic review paper is truly commendable, and I have no doubt that your work will be well received by the scientific community.

Author Response

Response to Reviewer 2 Comments

Comments and Suggestions for Authors

Your work on the historical course of brain tuberculosis is truly commendable. The thoroughness with which you have researched this complex subject is a testament to your dedication and expertise in the field. By illuminating the often overlooked historical context of this disease, your manuscript provides invaluable insights that enrich our understanding of tuberculosis as a whole. The choice of topics and the logical structure of your report make it a compelling and exciting read. Your efforts to cover a broad period from antiquity to the present demonstrate a great commitment to presenting a holistic view of this subject.
Some general comments:
1. Abstract: It would be beneficial to reword the abstract to align it more closely with the text, especially with regard to the goal and conclusion.
2. Introduction:  Consider explicitly stating in the introduction the purpose of the article, the nature of the study conducted, and its overall significance to provide clear guidance to the reader.

  1. The TB treatment: Given the significant advances that have been made over the past five decades, I recommend that you update the manuscript with the most recent findings, particularly with respect to the treatment of multidrug-resistant forms TB. Including these advances in both the main text and Table 1 would increase the relevance and impact of your work.

I hope you find these comments helpful in further refining your manuscript. Your commitment to submitting a comprehensive and academic review paper is truly commendable, and I have no doubt that your work will be well received by the scientific community.

Response 2:

The authors would like to thank the reviewer for his sincere words of appreciation. We feel honored and appreciate your support and the patience with which you reviewed our article.

  • we added a paragraph at the end of the abstract:

“Understanding the history of treatment methods against this disease, from empirical and primitive ones to the emergence of new drugs used in multi-drug-resistant tuberculosis, makes us once again realize the significant contribution of science and medicine in treating a disease that was considered incurable not long ago.”

  • we added a paragraph in the introduction to better define the purpose of our work:

“The purpose of this article is to make an exhaustive journey through history, from the earliest records in antiquity regarding TB to the way treatment methods against this dis-ease have evolved: from primitive methods to the emergence of newly drugs administered in multidrug-resistant TB. It is worth noting that these treatment methods have evolved in parallel with humanity's understanding of the disease's origin, transmission, and clinical and pathological manifestations. Let's enjoy this journey!”

  • we have added a consistent paragraph about the latest discoveries in the field of anti-tuberculosis therapy. The key moments were also added to the table.

“In recent decades, drug-resistant TB (DR-TB) has emerged as the main challenge for the WHO Global TB Programme, as it increases the risk of disease relapse, treatment fail-ure, and death [101]. Multidrug-resistant tuberculosis (MDR-TB) is defined as the appear-ance of MTB strains resistant to at least rifampicin and isoniazid, making these patients practically incurable using standard first-line TB drugs [102, 103]. Additionally, in 2006, a new category named extensively drug-resistant (XDR) TB emerged, referring to MDR-TB strains that are resistant to fluoroquinolones and second-line injectable drugs [102]. MDR-TB has a high mortality rate, and due to the spread of this new form, there is a risk of tu-berculosis becoming an incurable disease. These concerns have led to numerous studies in recent years that have evaluated the risk factors for MDR-TB, such as lung cavity, previ-ously diagnosed TB, previous anti-TB therapy, and sputum acid-fast bacilli smear positivi-ty [104-106], which are risk factors independent of the global context. Considering these aspects, special attention should be given to these categories of patients.

The constant and ongoing spread of MDR-TB represents a challenge and an urgent situation for TB control globally [102]. In this regard, scientists have made tremendous efforts to identify new anti-tuberculosis drugs.

In 2005, Andries et al. [107] discovered bedaquiline, which was later approved by the US FDA for the treatment of MDR-TB in 2012. Subsequently, delamanid emerged, which proved its effectiveness first in vivo and in vitro preclinical studies [108, 109] and later in multinational clinical trials [110].

In 2013, delamanid was approved by the European Medicines Agency. Another sig-nificant moment in TB treatment was in 2019 when the first 6-month regimen was ap-proved for the treatment of MDR and XDR TB, consisting of bedaquiline, pretomanid, and linezolid [111]. Recently, the Working Group on New TB Drugs is trying to accelerate the discovery and development of new drugs for TB treatment [112].”

Your words of appreciation encourage us even more to realize a new review on this topic that has become relevant again in the last years.

Thany you once again for the extremely valuable comments.